# Trend and epidemiology of suicide attempts by self-poisoning among Egyptians

**Zeinab A. Kasemy**[1]*, **Asmaa Fady Sharif**[2,3], **Safaa Abdelzaher Amin**[4], **Manar Maher Fayed**[2], **Dalia E. Desouky**[1], **Amal A. Salama**[5], **Hanaa Mohammad Abo Shereda**[6], **Nehad B. Abdel-Aaty**[1]

**1** Department of Public Health and Community Medicine, Faculty of Medicine, Menoufia University, Menoufia, Egypt, **2** Department of Forensic Medicine and Clinical Toxicology, Faculty of Medicine, Tanta University, Tanta, Egypt, **3** Department of Clinical Sciences, College of Medicine, Dar Al Uloom University, Riyadh, Saudi Arabia, **4** Department of Forensic medicine and clinical toxicology, Faculty of Medicine, Menoufia University, Menoufia, Egypt, **5** Department of Family medicine, Faculty of Medicine, Menoufia University, Menoufia, Egypt, **6** Department of Psychiatric and Mental Health Nursing, Faculty of Nursing, Menoufia University, Menoufia, Egypt

* zeinabkasemy@gmail.com, zeinabkasemy@yahoo.com, zeinab.kasemy@med.menofia.edu.eg

**Data Availability Statement:** All relevant data are within the article.

**Funding:** The author(s) received no specific funding for this work.

## Abstract

Suicide attempts by self-poisoning have become a critical health problem. This study aimed to investigate the trend, incidence, and the associated risk factors of suicide attempts by self-poisoning. A total of 7398 Egyptian patients were analyzed. The trend of suicide attempts by self-poisoning was analyzed using 6745 patients over four registry years from January 1, 2016, to January 1, 2020. Then, the associated risk factors behind attempted suicide by self-poisoning from January 1, 2019, to January 1, 2020, were assessed using 2523 suicide attempters by self-poisoning, 201 fatalities by self-poisoning, and another 653 survivors of accidental poisoning. Results showed a rising trend of suicide attempts by self-poisoning over the studied years. The incidence of suicide attempts through deliberate self-poisoning represented 26.63/1,000 (CI95%: 25.63–27.86) to the admitted patients and 26.10/100,000 (CI95%: 25.10–27.14) to the regional population. The death rate due to suicide attempts by self-poisoning was 2.08/100,000 (1.90–2.49). The case fatality rate and the proportionate mortality rate for suicide by self-poisoning were 7.38% (CI95%: 6.45–8.42) and 14.11% (CI95%: 12.4–16.0) respectively. Multivariate analysis revealed that attempted suicide by self-poisoning was predicted among patients aged <25 or 25–40 years old (OR = 27.49, CI95%: 15.28–49.64 and OR = 59.42, CI95%: 32.76–107.77 respectively), those of low or moderate socioeconomic status (OR = 35.03, CI95%: 21.32–57.56 and OR = 14.11, CI95%: 10.86–18.43 respectively), students (OR = 2.91, CI95%: 1.57–5.43) and those living in rural residency (OR = 4.12, CI95%: 3.27–5.19). Suicide attempts by self-poisoning exhibited an incremental rise across time which raises a serious concern. Efforts should be directed to overcome the mentioned risk factors triggering suicide attempts by self-poisoning.

**Competing interests:** The authors have declared that no competing interests exist.

## Introduction

Suicide, either attempted or completed, is a significant problem that affects young people and adolescents. WHO states that suicide is the worldwide 2nd leading cause of death among people aged 15–29 year [1]. Although WHO states an overall decline in the global age-standardized suicide rates in the suicide worldwide 2019 report (except in some regions, for example, the Americas), these rates vary between countries from less than two deaths by suicide /100,000 to more than 80/100,000 [2]. Most deaths by suicide occurred in developing countries with low incomes [3].

Regrettably, the suicide rate in the Middle East region doubled from 1990 to 2015 [4]. Although the rate in Upper Egypt ranged from 0.6 to 0.8 per 100,000 [8], these figures include only the successful suicides. Para-suicide, which refers to failed attempts of non-fatal self-harm, is 30 times more common than completed suicide [5, 6].

Suicide methods vary worldwide based on socio-cultural backgrounds and differences in legislation between countries which not only affect the suicide rate but also suicide methods [5, 6]. Self-poisoning, firearms, and hanging are the most commonly used [7]. In the United Kingdom, suicidal self-poisoning represents approximately 25% of total suicides, while the United States considers it the most commonly used way of attempting suicide among youths [8, 9].The situation worsens in the developing world with a steady increase in deaths from suicide by self-poisoning [10]. In Egypt, there is a significant 38.4% increase in suicide attempts by self-poisoning among youths [11].

Attempted self-poisoning has replaced the traditional methods due to easier access to drugs and chemicals. Pesticides alone account for one-quarter of global suicide rates [12]. In developed countries, drugs used in suicidal attempts by self-poisoning include psychotropic drugs, analgesics, antihistamines, antidepressants, psychoactive drugs, and sedative-hypnotics [9, 13]. The situation is somewhat different in developing countries because pesticides are the most widely used method of suicide. However, central nervous system affecting drugs, and analgesics are commonly used agents [9, 13]. On the other side, pesticides were considered the chief materials used for attempting suicide by self-poisoning in Egypt. The residence and age of patients attempting suicide influence the type of material they use to attempt suicide. Though youth victims prefer xenobiotics, pesticides are commonly used by older subjects living in rural areas [14]. Previous studies investigating attempted suicide by self-poisoning in our region are primarily insufficient and are based on previous numbers that may be misleading or inexact [15]. This work aimed to assess the trend, incidence, and risk factors associated with suicide attempts by self-poisoning among Egyptians by examining cases admitted to approved poison control centers.

## Materials and methods

### Study design and setting

An analytical study was conducted on patients presented Tanta and Menoufia Universities Poison Control Centers that serve two large Egyptian governorates with a population of at least 10,000,000 people living in an area of more than 4485.03 square kilometers. Medical records of patients presented over four registry years from January 1st, 2016, to January 1st, 2020, were reviewed. Those diagnosed with suicidal attempts by self-poisoning were analyzed to assess the trend of suicide. Then, a comprehensive analysis of attempted suicide from January 1, 2019, to January 1, 2020, compared to survivors of accidental poisoning during the same year was conducted to assess the associated risk factors.

## Subjects and sampling

A total of 7,398 Egyptian patients were recruited. Among them, 6745 participated in trend analysis. To assess the risk factors associated with attempted suicide by self-poisoning, 3377 patients were analyzed including 2523 suicide attempters by self-poisoning, 201 fatalities by self-poisoning and a further 653 accidental poisoning survivors (representing the last year admitted patients January 1, 2019-January 1, 2020). Demographic data of suicide attempters by self-poisoning were compared with demographics of fatalities by self-poisoning and accidental poisoning survivors. Moreover, the attempted suicide by self-poisoning was compared with fatalities by self-poisoning in terms of the causative agent exposure and the reported reasons. Furthermore, we calculated the death rates, case fatality rates, and Proportionate mortality rate and mortality rates. **Fig 1** describes the distribution of the studied patients according to the manner of exposure and data analysis.

## Inclusion criteria

All patients admitted to poison control centers with complete medical records were c eligible. Drug poisoning was established as per the International Classification of Diseases. Egyptian poison control centers are the only governmental authorities delegated to receive admissions of acutely intoxicated patients and patients receiving toxins or chemicals regardless of exposure circumstances, provided that they were presented alive. Deaths before referral are neither admitted nor counted in the database. All confirmed cases are documented in the database, while undiagnosed query cases were reported under the query category. The exposure pattern was estimated based on the history provided by adult patients or their caregivers in the case of minors or mentally disabled patients. In addition, the manner of self-poisoning could be concluded easily as some patients were reported in the database for previous suicide attempts. Other rare cases bear the signs of previous suicide attempts in other ways, e.g., cut the wrists etc.

## Exclusion criteria

It included patients with suspected addiction, iatrogenic intoxication, and intoxication due to chronic drug use, as well as patients who had been discharged from the hospital against medical advice or who had spent < 24 hours with an uncertain fate.

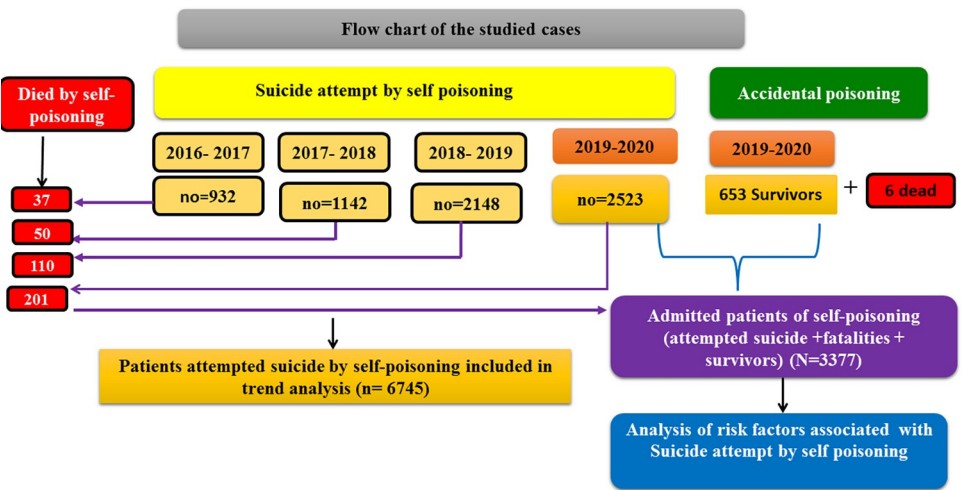

**Fig 1. Flow charts of the studied cases according to the manner of exposure, and the analysis they were included in.**

## Data collection tools

Medical records of patients admitted from January 1, 2016, to January 1, 2020, were extensively reviewed. From these records, we extracted the number of cases of attempted suicide per year which was used in trend analysis. Then, for January 2019-January 2020, we recruited cases of suicidal attempt by self-poisoning, fatalities by self-poisoning and cases of accidental poisoning. A predesigned case report form was distributed for both mentioned centers. Once upon admitting the patients, clinical and toxicological assessment was conducted. The manner of poisoning was determined (suicide attempt, or accidental). Furthermore, the type of chemical was determined from the history reported by the patients themselves or companions or in a few cases some patients brought the medicine package used. If necessary, clinical and/or laboratory examination were used to confirm the diagnosis [16].

A team consisting of a toxicologist, a psychiatrist, and a nurse was assigned to each suicide attempt. Demographic data included age, gender, residence, occupation, and socioeconomic standard. Patients were asked to answer questions about the triggers for the suicide attempt (financial issues, family conflicts, emotional issues, or bullying). Patients who could not choose among these factors were considered unknown risk factors.

The socioeconomic standard was categorized into high, medium, or low [17]. Subsequently, patients underwent a psychiatric evaluation according to Diagnostic and Statistical Manual of Mental Disorders, Fourth Edition criteria [18].

## Compliance with ethical standards

The current study began after obtaining approvals from both centers research ethics committees (ID: 2/2021COM and No.: 34244/11/20). Patient confidentiality was maintained after the Helsinki Declaration. The medical records were completely anonymous. The above-mentioned research ethical committees have waived the requirement for informed consent.

## Statistical analysis

Data were analyzed using SPSS 28. Qualitative data were presented as numbers and percentages. Graphing of the trend over four registry years was conducted using Microsoft Excel 2010. A Chi-square test ($X^2$) was used to compare qualitative variables. Binary logistic regression analysis was implemented to determine the associated risk factors for attempting suicide by self-poisoning. All significant variables in the univariate logistic regression were subjected to multivariate logistic regression to detect the riskiest factors for attempted suicide [19]. A p value < 0.05 was considered significant. The incidence rate was estimated per 1,000 patients admitted and 100,000 population per governorate and calculated by dividing the number of events by the number of patients admitted or the number of regional populations derived from official population data [20]. Case fatality rate was calculated as the number of self-poisoning-specific deaths among the incident cases divided by total number of incident cases. Proportionate mortality rate was calculated as the number of deaths due to self-poisoning divided by deaths from all causes.

## Results

### Trend of suicide attempts by self-poisoning during four years of registration data

The number and incidence of suicide attempts by self-poisoning for the years 2016–2017, 2017–2018, 2018–2019, and 2019–2020 are shown in **(Figs 2–4).** The number of suicide attempts by self-poisoning increased from 932 in 2016 to 2523 by January 2020. The current

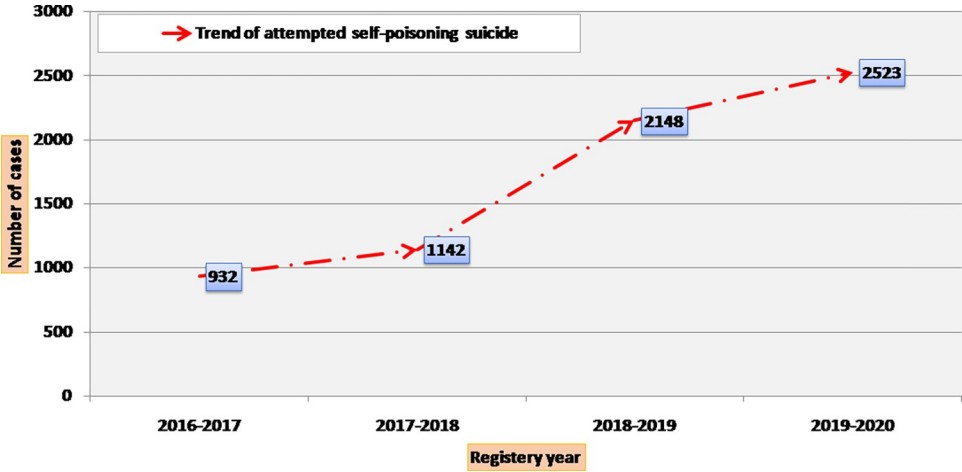

**Fig 2. Trend of attempted suicide by self-poisoning (number of cases during 4-year registry data).**

study showed an apparent increase in cases of suicidal self-poisoning and attempted suicide by self-poisoning during the four years studied.

## The incidence of suicide attempts by self-poisoning and deaths during the year 2019–2020

From January 1, 2019, to January 1, 2020, the total number of patients with suicidal self-poisoning was 2724 (n = 2523 attempts + 201 completed). The total number of accidental poisoning cases was 659 (n = 653 survivors + 6 deaths)). Suicide attempters by self-poisoning were 79.4% compared to 20.6% for accidental poisoning patients with an incidence rate of 26.63/1,000 admissions (CI95%: 25.63–27.86) and 26.10/100,000 population (CI95%: 25.10–27.14). The suicide rate by self-poisoning is 2.1/100,000 (1.9–2.5). The case-fatality rate and proportionate mortality rate from suicide by self-poisoning were 7.4% (CI95%: 6.5–8.4) and 14.1% (CI95%: 12.4–16.0), respectively (**Table 1**).

Out of the 2724 cases of attempted and successful suicide by self-poisoning during the year 2019–2020, females, students, individuals aged less than 25 years old, those living in urban

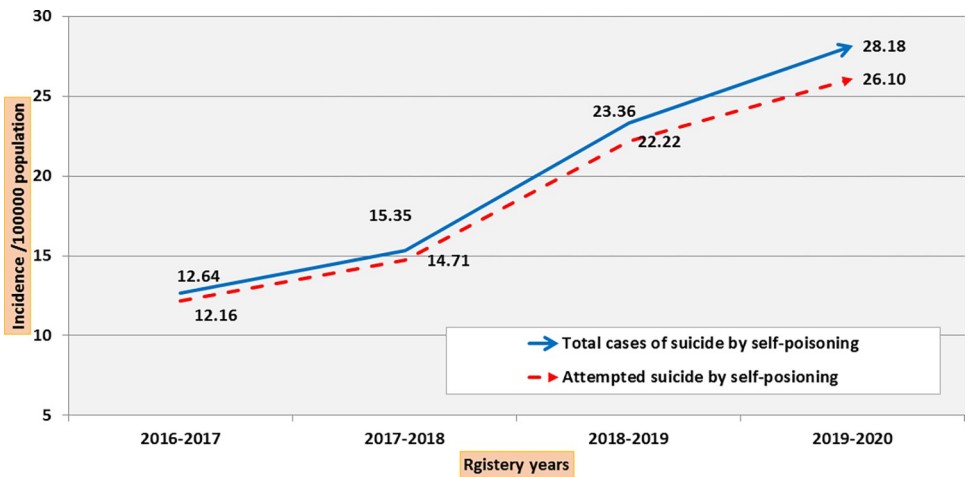

**Fig 3. Trend of attempted suicide by self-poisoning (incidence /100000 population during 4-year registry data).**
The total number of suicides by self-poisoning reflects the sum of attempted and successful suicide by self-poisoning.

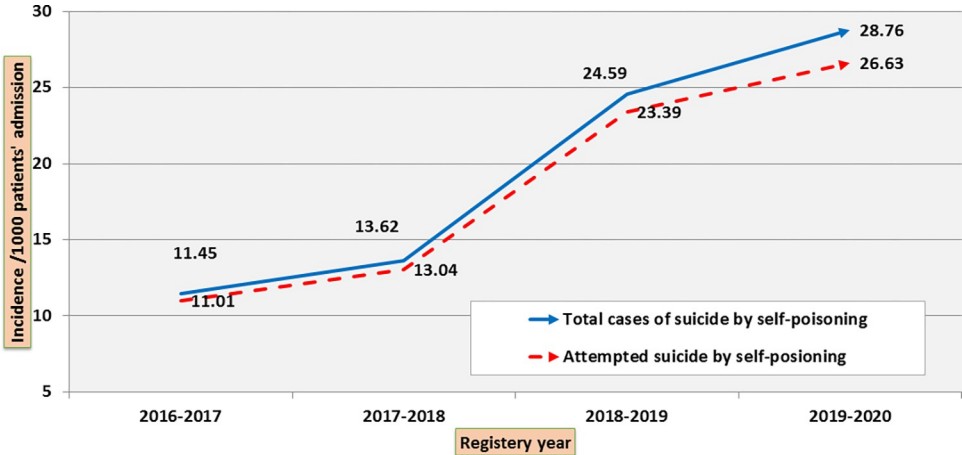

**Fig 4. Trend of attempted suicide by self-poisoning (incidence /1000 admitted patients during 4-year registry data).** The total number of suicides by self-poisoning reflects the sum of attempted and successful suicide by self-poisoning.

areas (62.7%), those suffering from family disputes (58.2%), or psychological disorders (37.3%) had increased odds of death and were significantly at risk of death from suicide by self-poisoning more than others. Furthermore, Aluminum phosphide was the leading cause of suicidal death by self-poisoning (94.5%). The most widely used way for attempting suicide by self-

**Table 1. Incidence and associated deaths rates of attempted suicide by self-poisoning during the year Jan 2019–Jan 2020.**

|  | No | (CI95%) |
|---|---|---|
| **Poisoning type** |  |  |
| • Suicide by self-poisoning (attempted +completed) | 2724 | 80.5% (80.4–80.6) |
| • Accidental Poisoning (survivors +dead) | 659 | 19.5% (19.4–19.5) |
| **Living** |  |  |
| • Suicide by self-poisoning attempters | 2523 | 79.4% (77.9–80.8) |
| • Accidental Poisoning survivors | 653 | 20.6% (19.1–22.0) |
| **Death** |  |  |
| • Completed suicide by Self-poisoning | 201 | 97.1% (93.8–98.9) |
| • Dead by Accidental poisoning | 6 | 2.9% (1.1–6.2) |
|  | **Incidence rate** | **CI95%** |
| **Incidence rate of poisoning in relation to all admitted patients** |  |  |
| • Suicide attempt by Self-poisoning (n = 2523) | 26.63/1000 | 25.63–27.86 |
| • Accidental Poisoning (n = 653) | 06.89/100 | 6.38–7.44 |
| **Incidence rate of suicide in relation to all regional population** |  |  |
| • Suicide attempt by Self-poisoning(n = 2523) | 26.10/100000 | 25.10–27.14 |
| • Accidental Poisoning(n = 653) | 6.76/100000 | 6.26–7.29 |
| **Self-Poisoning suicide death rate (n = 201/2724)** | 2.08/100000 | 1.90–2.49 |
| **Case fatality rate[a]** |  |  |
| • Suicide by Self-poisoning (n = 201/2724) | 7.38% | 6.45–8.42 |
| • Accidental Poisoning (n = 6/659) | 0.91% | 0.41–2.01 |
| • Total (Suicide by Self-poisoning + accidental poisoning) | 6.12% | 5.36–6.98 |
| **Proportionate mortality rate[b] (n = 201/1424)** | 14.11% | 12.43–16.0 |

a. Case fatality rate was calculated as the number of self-poisoning-specific deaths among the incident cases divided by total number of incident cases.

b. Proportionate mortality rate was calculated as the number of deaths due to self-poisoning divided by deaths from all causes.

poisoning was pesticides (38.7%), then aluminum phosphide (29.3%). Drugs constituted about (18.5%) and included central nervous system (CNS) depressants (40.7%), cardiovascular and xanthine derivatives (27.4%), antipsychotics (21.8%), and antidepressants (20.9%). Family conflicts (30.8%), financial problems (23.6%) and psychological problems (21.4%) constituted the main conveyed reasons for attempting suicide by self-poisoning (**Table 2**).

## Univariate analysis of risk factors associated with suicide attempts by self-poisoning

Univariate analysis for risk factors associated with attempted suicide by self-poisoning were age ≤25 years and 25–40 years old [OR = 6.97, CI95%: 5.29–9.20 and OR = 1.36, CI95%:

**Table 2. Distribution of attempted and successful suicide by self-poisoning (n = 2724) regarding the demographic data, causative agent exposure and reported reasons during the year 2019–2020.**

| | Groups | | | | p- value | OR [CI95%] |
|---|---|---|---|---|---|---|
| | Suicide by self-poisoning (n = 201) | | Self-poisoning attempters (n = 2523) | | | |
| | no | % | no | % | | |
| **Age (years)** | | | | | <0.001* | |
| ≤25 | 185 | 92.0 | 1196 | 47.4 | | 12.83[7.65–21.51] |
| >25 | 16 | 8.0 | 1327 | 52.6 | | 1.0 |
| **Gender** | | | | | <0.001* | |
| Male | 41 | 20.4 | 858 | 34.0 | | 1.0 [1.41–2.86] |
| Female | 160 | 79.6 | 1665 | 66 | | 2.01 |
| **Residence** | | | | | <0.001* | |
| Rural | 75 | 37.3 | 1533 | 60.8 | | 1.0 |
| Urban | 126 | 62.7 | 990 | 39.2 | | 2.60[1.93–3.50] |
| **Occupation** | | | | | | |
| Working | 2 | 1.0 | 674 | 26.7 | <0.001* | 1.0 |
| Not working | 42 | 20.9 | 903 | 35.8 | <0.001* | 55.93[13.82–226.40] |
| A student | 157 | 78.1 | 946 | 37.5 | | 15.67[3.78–64.98] |
| **SES** | | | | | | |
| Low | 71 | 35.3 | 1750 | 69.4 | 0.092 | 0.42[0.14–1.19] |
| Medium | 126 | 62.7 | 732 | 29.0 | 0.280 | 1.76[0.62–5.01] |
| High | 4 | 2.0 | 41 | 1.6 | | 1.0 |
| **Reported causes** | | | | | <0.001* | - |
| Family disputes | 117 | 58.2 | 776 | 30.8 | | |
| Psychological disorders | 75 | 37.3 | 541 | 21.4 | | |
| Financial issues | 2 | 1.0 | 596 | 23.6 | | |
| Emotional issues | 7 | 3.5 | 340 | 13.5 | | |
| Bullying | 0 | 0.0 | 156 | 6.2 | | |
| Unknown | 0 | 0.0 | 114 | 4.5 | | |
| **Causative Agent** | | | | | | |
| Pesticide | 4 | 2.0 | 976 | 38.7 | <0.001* | - |
| Aluminum phosphide | 190 | 94.5 | 740 | 29.3 | | |
| Zinc phosphide | 7 | 3.5 | 257 | 10.2 | | |
| Hydrocarbons | 0 | 0.0 | 83 | 3.3 | | |
| Drugs | 0 | 0.0 | 467 | 18.5 | | |

*: significant SES: socioeconomic standard

**Table 3. Univariate analysis of risk factors associated with attempted suicide by self-poisoning versus accidental poisoning during the year 2019–2020.**

| | Poisoning | | | | | | p- value | OR [CI95%] |
|---|---|---|---|---|---|---|---|---|
| | Self-poisoning attempters (n = 2523) | | Accidental poisoning survivors (n = 653) | | Total (n = 3176) | | | |
| | no | % | no | % | no | % | | |
| **Age (years)** | | | | | | | | |
| ≤25y | 1196 | 47.4 | 90 | 13.8 | 1286 | 40.5 | <0.001* | 6.97 [5.29–9.20] |
| >25-40y | 965 | 38.2 | 373 | 57.1 | 1338 | 42.1 | 0.001* | 1.36 [1.10–1.68] |
| >40y | 362 | 14.3 | 190 | 29.1 | 552 | 17.4 | - | 1.0 |
| **Gender** | | | | | | | | |
| Male | 858 | 34.0 | 334 | 51.1 | 1192 | 37.5 | <0.001* | 1.0 |
| Female | 1665 | 66.0 | 319 | 48.9 | 1984 | 62.5 | | 2.03[1.71–2.42] |
| **Residence** | | | | | | | <0.001* | |
| Rural | 1533 | 60.8 | 246 | 37.7 | 1779 | 52.6 | | 2.56 [2.15–3.06] |
| Urban | 990 | 39.2 | 407 | 62.3 | 1397 | 47.4 | | 1.0 |
| **Occupation** | | | | | | | | |
| Working | 674 | 26.7 | 303 | 46.0 | 977 | 30.8 | - | 1.0 |
| Not working | 903 | 35.8 | 266 | 40.4 | 1363 | 36.6 | <0.001* | 1.59 [1.31–1.93] |
| A student | 946 | 37.5 | 90 | 13.7 | 1036 | 32.6 | <0.001* | 5.49 [4.26–7.08] |
| **SES** | | | | | | | | |
| Low | 1750 | 69.4 | 197 | 30.2 | 1947 | 61.3 | <0.001* | 13.43 [8.82–20.47] |
| Medium | 732 | 29.0 | 394 | 60.3 | 1126 | 35.5 | <0.001* | 2.81 [1.86–4.25] |
| High | 41 | 1.6 | 62 | 9.5 | 103 | 3.2 | - | 1.0 |

*: significant SES: socioeconomic standard

1.10–1.68 respectively], being a female [OR = 2.03, CI95%: 1.71–2.42], a student [OR = 5.49, CI95%: 4.26–7.08], not working individual (OR = 1.59, CI95%: 1.31–1.93), being a rural resident [OR = 2.56, CI95%: 2.15–3.06] and those with low and moderate socioeconomic status [OR = 13.43, CI95%: 8.82–20.47 and OR = 2.81, CI95%: 1.86–4.25 respectively] (**Table 3**).

## Multivariate analysis of risk factors associated with attempting suicide by self-poisoning

Multivariate analysis revealed that attempted suicide by self-poisoning was predicted among patients aged <25 or 25–40 years old (OR = 27.49, CI95%: 15.28–49.64 and OR = 59.42, CI95%: 32.76–107.77 respectively), those of low or moderate socioeconomic status ((OR = 35.03, CI95%: 21.32–57.56 and OR = 14.11, CI95%: 10.86–18.43 respectively), students (OR = 2.91,CI95%: 1.57–5.43) and those living in rural residency (OR = 4.12, CI95%: 3.27–5.19). The logistic regression model designed to assess predictors of suicide attempts was statistically significant (p < 0.001 justifying 43.7% (NagelkerkeR2) of the variance in suicide and appropriately classified 86.3% of patients (**Table 4**).

## Discussion

This study showed that (79.4%) of the patients included in the study attempted suicide by self-poisoning. The rate of suicide attempts by self-poisoning in the general population was 26.10/100,000. Moreover, it was elaborated in the current study that 1196 out of 2523 suicidal attempters during the year 2019–2020, were less than 25 years, representing about (47.4%) of total suicide attempts by self-poisoning. Similarly, a dramatic increase in suicide attempts by

**Table 4. Multivariate analysis of risk factors associated with attempted suicide by self-poisoning using Binary logistic regression.**

|  | P value | OR | CI95% | |
|---|---|---|---|---|
|  |  |  | Lower | Upper |
| **Age** | <0.001* |  |  |  |
| Age: 25–40 | <0.001* | 59.42 | 32.76 | 107.77 |
| Age (<25) | <0.001* | 27.49 | 15.28 | 49.46 |
| **Sex (Female)** | 0.064 | 1.26 | 0.99 | 1.60 |
| **Residence (Rural)** | <0.001* | 4.12 | 3.27 | 5.19 |
| **Socioeconomic standard** | <0.001* |  |  |  |
| Moderate | <0.001* | 14.11 | 10.86 | 18.34 |
| Low | <0.001* | 35.03 | 21.32 | 57.56 |
| **Occupation** | 0.002* |  |  |  |
| Not working | 0.719 | 1.05 | 0.80 | 1.38 |
| Students | <0.001* | 2.91 | 1.57 | 5.43 |

*: significant

self-poisoning was reported, especially in children aged less than 19 over the last 10 years [21]. These numbers are much higher than those reported earlier in 2004, which reported that about 2.9% of teens who attempted suicide needed emergency admission. Though this rise reflects the catastrophic surge of this serious problem, part of the gap in numbers is attributable to the sampling difference between the different studies. Our study targets only those admitted to poison centers while the mentioned study, in 2004, calculated this rate considering the general population [22].

Nevertheless, Suicides by self-poisoning/1,000 admissions were significantly increased over a four-year period. Suicides by self-poisoning reached 28.76/1000 populations including attempted and successful suicides in 2019–2020, reflecting the severity and evolution of this problem. It has been reported that the suicide rate has increased by 16% over the past decade in the United States [23]. Studies in Australia, the United Kingdom, Italy and Brazil have reported an increasing trend, raising global concerns [24–27].

The current study reported that the death rate due to suicide by self-poisoning was 2.08/100,000 during 2019. The rate ranged between 0.47 and 2.41 during the past two decades [28–30] The rate changes from study to study and may vary in different countries over time, and even within the same country. This difference can be illustrated by the difference in the culture of the population studied, the size of the carefully studied environmental area, the health care system facilities, and the diversity in socio-demographic conditions. Also, preliminary screening may occur, which may reduce the proper number of suicide attempts from one area to another [31–34].

The most common method used to attempt suicide by self-poisoning in the current study was pesticides. The main drugs used were central nervous system depressants while aluminum phosphide was the leading cause of suicidal deaths due to self-poisoning. Similarly, In Sri Lanka, nearly half of self-poisoning cases are attributed to pesticides [35]. These findings are consistent with other studies in different contexts [12, 29, 36]. An earlier study in Egypt warranted the increased deaths due to self-poisoning using the aluminum phosphide [14]. These products' easy availability and low cost (pesticides) may explain their widespread use. On the contrary, in developed countries, drugs that act in the central nervous system are the most common means of attempting suicide by self-poisoning [37]. Differences between countries can be attributed to differences in socioeconomic conditions and access to medicines [38].

The current study investigated some motives for attempting suicide by self-poisoning, such as family disputes and psychological disorders. These findings coincide with previous studies suggesting that attempted suicide by self-poisoning may be related to recent family arguments or psychiatric disorders, in which emotional release or coping strategies can occur [30, 39, 40]. The association between mental disorders and the suicide attempt is consistent with additional studies in different contexts [6, 41]. Exacerbation of depression in the population, which may be exacerbated by societal problems such as unemployment, may lead to attempted suicide by self-poisoning. Conflicts, relationship failures, and examination failures are found in developing countries [42].

Furthermore, the stigma associated with suicide attempts can lower incident reporting. In our study, the causes of attempted suicide are unknown in about 5% of patients. This may be attributed to the tendency of patients' relatives to deny or refuse to acknowledge the attempted suicide. Some patients do not have families, and clinicians also face obstacles in evaluating and managing such cases [43, 44].

The present work reported that subjects under 25 were more likely to attempt suicide by self-poisoning than other elderly people. This may be related to the emotional distress and unemployment that young people face and the unfortunately and predictable overdose medications commonly consumed by this age group, and the responsibility of older adults to their families may hinder such suicidal thoughts [29, 36, 45].

Easy access to the internet, social media, and smart device usage among this age group can negatively impact mental health and ability to deal with various stressors. Egypt ranks high in internet use in the Middle East and shows high smartphone use among youth (90%) [46].

The current study described that female are at greater risk of attempting suicide by self-poisoning than males. Globally, self-harm is ranked second among females and third among males [40, 47]. A recent study in Egypt reported that two-thirds of suicide self-poisoning cases occurred among females [48]. In Australia, the number of females attempting suicide by self-poisoning was greater than that of males (3:1) [24]. Females tend to consume items already available in the home, while males tend to ingest or purchase an item kept outside [48].

Availability of the substance, especially pesticides, as well as hormonal disturbances and the habitual tendency of females to conflict with their guardians, are among the main reasons for attempting suicide by self-poisoning [49–51]. The burden of spinsterhood on females, along with low socioeconomic levels and low wages, paves the way for the spread of suicide attempts by female self-poisoning [48].

Qin et al. elaborated on the gender differences among patients of attempted suicide. Mental disorders, living in urban areas, without children, and having a family history of suicide increases the risk of suicide in females compared to males [41]. Toth et al., studied gender differences in suicide attempts as there was a gender factor in the presentation of depression among patients with interpersonal struggles. Females with interpersonal conflicts who attempted suicide by self-poisoning showed higher levels of depression than males [52].

Students and patients who did not work showed a significantly suicide attempts by self-poisoning. This finding has been reinforced elsewhere [53, 54]. Another study described even low-paid work as having a direct primary link to higher suicide rates. Moreover, this study suggested that an increase in salaries would lead to a rapid decrease in the suicide rate [55].

Our study showed that patients with low socioeconomic status and those living in rural areas were at significantly increased risk of attempting suicide. This result was agreed upon by Liu et al. It can be justified by the resulting anxiety, and the consequent illicit drug use associated with low income [56, 57]. However, this is in contrasts to Finkelstein et al., who reported conflicting results and stated that hospitalization due to suicidal overdose is more common among patients with high socioeconomic conditions [23]. Sun et al. found the rural-urban

discrepancy 2.5:1, indicating that the difference can be justified by differences in age, literacy, occupation and residency [57]. On the contrary, another study conducted in rural Sri Lanka reported a significant reduction in the suicide attempts by self-poisoning. The authors attribute this to the heavy dependence on agriculture and grazing for subsistence without the need for additional resources [58].

Preventing suicide by self-poisoning and all types of suicide is a public health responsibility to overcome the increasing trend in suicide rates, which affect individuals and families, communities, and society. Therefore, putting in place policies to reduce such a risk is urgent. Nurses, toxicologists, and public health professionals remain focused within the health care professional community on identifying risk factors for suicide and implementing these policies in practice. It is necessary to know the groups at high risk [59].

Nurses working in psychiatric liaison, CAMHS or primary care must have in-depth knowledge and skills to conduct comprehensive assessments. It is necessary to adopt approaches that address the individual and societal factors that motivate suicide. In societies where basic needs such as the emotional, financial and religious needs of its inhabitants are available, suicide and its consequences rarely occur.

## Strengths and limitations

Although the study was conducted in two Egyptian governorates only, it provided a recent report on the seriousness of the current suicide attempts through self-poisoning. Examining several issues such as follow-up of survivors to check whether cases were referred for psychological counseling, and whether there were any other hidden factors to avoid a recurrence of the suicide incident was out of our hands. This might be due to the sensitivity of this topic and the stigma associated with it within the victims' families.

## Conclusion

Suicide by self-poisoning is increasing in Egypt, which raises concern. Family disputes, financial and psychological issues were the main reasons for attempting suicide by self-poisoning. Students, those under the age of 40, those who live in rural areas and those of low and moderate socioeconomic standards are more likely to attempt suicide by self-poisoning. Attempted suicide by self-poisoning is a multifactorial problem that requires a deep understanding and cooperation among all interested parties including physicians, nurses, family, community, clergy and state leaders to provide support, health education and legislation.

## Author Contributions

**Conceptualization:** Zeinab A. Kasemy, Safaa Abdelzaher Amin, Manar Maher Fayed.

**Data curation:** Zeinab A. Kasemy, Asmaa Fady Sharif, Manar Maher Fayed.

**Investigation:** Safaa Abdelzaher Amin.

**Methodology:** Zeinab A. Kasemy, Safaa Abdelzaher Amin.

**Resources:** Safaa Abdelzaher Amin, Dalia E. Desouky, Hanaa Mohammad Abo Shereda.

**Software:** Zeinab A. Kasemy.

**Supervision:** Dalia E. Desouky.

**Validation:** Zeinab A. Kasemy.

**Writing – original draft:** Zeinab A. Kasemy, Asmaa Fady Sharif, Manar Maher Fayed, Amal A. Salama, Nehad B. Abdel-Aaty.

**Writing – review & editing:** Zeinab A. Kasemy, Asmaa Fady Sharif, Manar Maher Fayed, Nehad B. Abdel-Aaty.

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
