## [Decision Letter · Decision Letter 0]

1 Feb 2021

PONE-D-20-34054

Incidence, Distribution, and Associated Factors of Suicide in Two Egyptian Provinces

PLOS ONE

Dear Dr. Kasemy,

Thank you for submitting your manuscript to PLOS ONE. After careful consideration, we feel that it has merit but does not fully meet PLOS ONE’s publication criteria as it currently stands. Therefore, we invite you to submit a revised version of the manuscript that addresses the points raised during the review process.

We look forward to receiving your revised manuscript.

Kind regards,

Vincenzo De Luca

Academic Editor

PLOS ONE

Journal Requirements:

2. We note that your study included both retrospective and prospective elements. Please update the manuscript to include further details regarding the prospective element of your study. Please ensure you have specified the demographics and characteristics of the individuals you collected data from, the data collected as part of this study and how the results were used to inform the results described. Please also ensure you have clearly indicated how the data reported in Results section were collected. We also note that your study lists of the causes of suicide. Please ensure that you have specified how this was determined in your Methods section.

4. Thank you for including your ethics statement:  "Approval was obtained from local Institutional Review Boards (IRB) following the Declaration of Helsinki under ID. ".   

Reviewers' comments:

Reviewer's Responses to Questions

**Comments to the Author**

1. Is the manuscript technically sound, and do the data support the conclusions?

Reviewer #1: Partly

2. Has the statistical analysis been performed appropriately and rigorously? 

Reviewer #1: I Don't Know

3. Have the authors made all data underlying the findings in their manuscript fully available?

Reviewer #1: No

4. Is the manuscript presented in an intelligible fashion and written in standard English?

Reviewer #1: No

5. Review Comments to the Author

Reviewer #1: The topic of the paper is clinically relevant and could help to explain and understand the specif-ic circumstances of suicidality in Middle Eastern countries compared to other parts of the world.

The title is not sufficiently focussed more a short version of the abstract. The abstract is too long for an original paper and needs to be more structured based on the established format of the journal.

For an international readership it would be especially interesting to compare international trends to the situation in Egypt and the middle eastern region. I would suggest to take out gen-eral statements on suicide and refer to the literature and focus on describing the specific cir-cumstances and risk factors for suicidal behaviour and suicide in the region. That should follow a clear structure in the introduction, so that the reader always knows what the authors are re-ferring to. Critical factors like cultural aspects, religion, family structures and gender related expectations and risk factors are not described in a way, that allows the reader to be prepared for the results section. Also the fact, that suicidal behaviour and thoughts are highly stigma-tized and the suicide rate might be under reported would be important for the conclusion and discussion later on. Also suicide rates in specific populations like street entrenched youth or rural populations are not reflected or presented. So the reader learns very little about specific vulnerable populations and risk factors.

Also to study something is not a sufficient objective. I’m sure the authors had interests and ideas beyond that.

The method section again needs a clear structure like describing the study sample e.g how many patients in phase 1 and 2 etc., the evaluation methods, the timeframe and the expertise used and available to carry out the research and the detailed statistical procedures used. The governorates were not mentioned so the reader can not apply context knowledge. The termi-nology between self-poisoning and suicide is used in a confusing way. It should be clearly em-phasized what the authors are talking about.

The lack of information and structure makes it complicated to understand the results. What constitutes the assessment of poisoning, Who is doing that and based on what criteria it is classified as suicide. In case of the use of psychotropic substances often several substances are used at once but there is no mention of polysubstance use. That way the results section is ex-tremely complicated to follow and unclear in its conclusions. A straight forward presentation of results should be well structured and allowing the reader to understand every step of the as-sessment!

The discussion of the paper is including a lot of interesting and important considerations, which should already been mentioned in the introduction. Again it needs restructuring in order to be able to draw conclusions. The discussion is also not answering the question related to the in-crease in suicidal actions. Also there is a lack of comparison to the international development and the international literature. Is Egypt in a specific situation? Are there specific provinces which are worse off and why? Some comments are inconsistent:

they mentioned suicide but their focus on self-poisoning

They mentioned that the study was along 4 years and, in the discussion, said 2019-2020 only.

They stated in the results the total number of self-poisons was 3383 and in discussion only the suicidal number (2724). Needs clarification.

They did not name the governorates or give any background about them.

They showed that the rates of their centers are much higher than the Egyptian national number without any rationalization.

The explanation of the third paragraph as a difference in the culture only needs to be more explained.

The explanation of the gender differences in the fourth paragraph as by hormonal caus-es only are not logic.

Same for the fifth paragraph, hormonal reason is reported as an explanation for suicide in students!

In the eighth paragraph, I think these are risk factors for suicidal self-poisoning not causes of suicide.

Risk factors have weak explanations.

Based on that the conclusions are not really conclusions because there are no recommenda-tions or suggested consequences based on the results presented.

I recommend the authors to collaborate with native speakers use more visuals in order to increase readability.

6. PLOS authors have the option to publish the peer review history of their article (what does this mean?). If published, this will include your full peer review and any attached files.

Reviewer #1: No

---

## [Author Response · Author response to Decision Letter 0]

19 Feb 2021

Dear editor and reviewers, thanks so much for your great and endless effort. We hope to find the response satisfactory and admired

---

## [Decision Letter · Decision Letter 1]

24 Jan 2022

PONE-D-20-34054R1Incidence, Distribution, and Determinants of Suicide by Self-poisoning in Two Egyptian ProvincesPLOS ONE

Dear Dr. Zeinab A. Kasemy

Thank you for submitting your manuscript to PLOS ONE. After careful consideration, we feel that it has merit but does not fully meet PLOS ONE’s publication criteria as it currently stands. Therefore, we invite you to submit a revised version of the manuscript that addresses the points raised during the review process.

We look forward to receiving your revised manuscript.

Kind regards,

Saeed Ahmed, MD

Academic Editor

PLOS ONE

Reviewers' comments:

Reviewer's Responses to Questions

**Comments to the Author**

1. If the authors have adequately addressed your comments raised in a previous round of review and you feel that this manuscript is now acceptable for publication, you may indicate that here to bypass the “Comments to the Author” section, enter your conflict of interest statement in the “Confidential to Editor” section, and submit your "Accept" recommendation.

Reviewer #2: (No Response)

Reviewer #3: (No Response)

2. Is the manuscript technically sound, and do the data support the conclusions?

Reviewer #2: No

Reviewer #3: Yes

3. Has the statistical analysis been performed appropriately and rigorously? 

Reviewer #2: Yes

Reviewer #3: (No Response)

4. Have the authors made all data underlying the findings in their manuscript fully available?

Reviewer #2: Yes

Reviewer #3: (No Response)

5. Is the manuscript presented in an intelligible fashion and written in standard English?

Reviewer #2: No

Reviewer #3: (No Response)

6. Review Comments to the Author

Reviewer #2: 1. Lines 24 and 25 mention, “the first part was conducted retrospectively to show the trend of suicide attempts over 2016, 2017, and 2018” - So, the 4419 patients mentioned here – are those who attempted suicide or those who died of suicide? I recommend that the data regarding patients diagnosed with suicide by self-poisoning and suicide attempt by self-poisoning is delineated and changed across the article (see point 8 below) accordingly.

2. Line 60 – “Reports published in the US considered suicide by self-poisoning the most common method of suicide attempts among adolescents (66.5%)”. I recommend removing suicide by self-poisoning and replacing it with self-poisoning as the article refers to suicide attempts and not suicide.

3. Line 77 – The sentence does not clearly communicate the author’s thoughts to the reader. Can you clarify “drugs?” I recommend rewriting the sentence for clarity.

4. Line 83-87 – The sentence is too long and loses clarity. I recommend breaking it into two sentences.

5. Line 97 – Same question as point 1. Does 969, 1,192, and 2,258 refer to patients who died of suicide or are these the number of patients who attempted suicide. Based on this the diagnosis should reflect suicide by self-poisoning versus suicide attempt by self-poisoning

6. Line 101- Why was accidental poisoning included in the study? Is there a relevance to suicide which is the primary objective of the study?

7. The lines 129-146 under data collection tools are confusing to the reader. Do these lines refer to patients who attempted suicide by overdose? Clarifying point 1 above and rewriting this section is recommended.

8. Lines 169-171, 173, 175, 186 and table 1, 222, 247, 274, 288, 298, 307, 316 and 343 – does this data pertain to suicide attempts or to suicide? If it refers to suicide attempts, then the sentences need to be changed to reflect that.

9. Line 224 – the article refers to suicide attempts and not suicide as mentioned in this line. Recommend changing it for accuracy.

10. Line starting 337 ending in 339 is incomplete and I recommend rewriting it to convey accurate meaning.

Reviewer #3: The manuscript by Kasemy, et al studies the important topic of suicide.

Strengths of the paper:

- The title is appropriate for the content of the text.

- The abstract describe findings of the study as well as suggestions on what can be done to prevent this global public health issue.

- The introduction to the article is well laid out that includes prevalence, specific methods of suicide, and risk factors, both nationally and globally.

- Methods and results are clearly described. Tables included in the manuscript help quickly review the study outcomes.

- Discussion and conclusion further elaborate risk factors and differences in methods of choice to commit suicide in Egypt and in comparison to other countries (vis-a-vis developed and developing countries) and the possible reason for those differences.

Some weaknesses are:

- Not always easily readability of the text and makes it difficult to follow at times.

- In line 47, it is not clear if the authors are implying suicide being the leading cause of death in children and adolescents or suicide deaths by numbers in this age group.

- In line 48, the authors report a steady increase in suicide rates globally (WHO 2001 data). However, new data in the Suicide worldwide 2019 report by WHO (published in 2021) states overall the global age-standardized suicide rate is somewhat in decline (except some regions, for example, the Americas). Ref: https://www.who.int/publications/i/item/9789240026643

- Authors rightly report stigma surrounding the suicide. In line 303, the authors suggest gender differences in suicide rates to hormonal or interpersonal conflict. The full comprehension of the gender paradox in suicidal behavior requires further research. Suggestion for authors to review studies on gender differences in suicide and other possible risk factors including but not limited to psychiatric disorders.

Ref: Qin, P., Agerbo, E., Westergård-Nielsen, N., Eriksson, T., & Mortensen, P. (2000). Gender differences in risk factors for suicide in Denmark. British Journal of Psychiatry, 177(6), 546-550. doi:10.1192/bjp.177.6.546. Tóth MD, Ádám S, Birkás E, Székely A, Stauder A, Purebl G. Gender differences in deliberate self-poisoning in Hungary: analyzing the effect of precipitating factors and their relation to depression. Crisis. 2014;35(3):145-53. doi: 10.1027/0227-5910/a000245. PMID: 24491825.

- In line 293, the authors report risk factors in less than 25 years old population. It is unclear if authors are implying social media as the monster or a specific online site named the monster.

- Lines 296, 300, 303 have missing references.

7. PLOS authors have the option to publish the peer review history of their article (what does this mean?). If published, this will include your full peer review and any attached files.

Reviewer #2: No

Reviewer #3: No

---

## [Author Response · Author response to Decision Letter 1]

5 Feb 2022

Based on your valuable comments, we have changed the article starting from the title till its end. The authors extend their appreciation for the reviewers for their great efforts in improving the quality of the manuscript. We found all proposed comments valuable and enriching the article. We hope we could fix the modifications meeting your expectations. Based on your valuable and comments, we changed the title and delineated the attempted from those died by suicide self-poisoning hoping to find satisfactory and clearly presented. All changes are found in documents of response to reviewer's comments and manuscript either with tack changes or without. I hope you find it well written and to the point. We did our best.

---

## [Decision Letter · Decision Letter 2]

11 Apr 2022

PONE-D-20-34054R2Trend and Epidemiology of Suicide attempts by Self-poisoning among EgyptiansPLOS ONE

Dear Dr. Kasemy,

Thank you for submitting your manuscript to PLOS ONE. After careful consideration, we feel that it has merit but does not fully meet PLOS ONE’s publication criteria as it currently stands. Therefore, we invite you to submit a revised version of the manuscript that addresses the points raised during the review process.

We look forward to receiving your revised manuscript.

Kind regards,

Saeed Ahmed, MD

Academic Editor

PLOS ONE

**Comments to the Author**

1. If the authors have adequately addressed your comments raised in a previous round of review and you feel that this manuscript is now acceptable for publication, you may indicate that here to bypass the “Comments to the Author” section, enter your conflict of interest statement in the “Confidential to Editor” section, and submit your "Accept" recommendation.

Reviewer #3: (No Response)

Reviewer #4: (No Response)

2. Is the manuscript technically sound, and do the data support the conclusions?

Reviewer #3: Yes

Reviewer #4: Partly

3. Has the statistical analysis been performed appropriately and rigorously? 

Reviewer #3: (No Response)

Reviewer #4: No

4. Have the authors made all data underlying the findings in their manuscript fully available?

Reviewer #3: (No Response)

Reviewer #4: Yes

5. Is the manuscript presented in an intelligible fashion and written in standard English?

Reviewer #3: No

Reviewer #4: Yes

6. Review Comments to the Author

Reviewer #3: The authors have adequately addressed comments raised in a previous round of review and feel that this manuscript is now acceptable for publication except for multiple grammatical errors. I encourage authors to fix these before publication.

Suggestion for authors: can use online freely available software like Grammarly, etc.

Reviewer #4: The authors present results from a study of trends in suicide and attempted suicide rates in Egypt over a 4 year period as well as risk factors for attempted suicide in 2019. Authors find that rates are increasing over time, that family, psychological issues, and financial issues are common reasons for attempted suicide, and that age, residential location, working status, and SES are risk factors for attempted suicide. The manuscript will be strengthened if the authors consider the following points:

1. Abstract: authors present results from the univariate logistic regression models. However, the multivariate model is more appropriate for final results, since this accounts for other variables included in the model.

2. In the subjects and sampling section of the Methods, authors state that fatalities were excluded from the analyses of risk factors. Authors should provide additional justification for this decision. Some of the information collected on the attempts are likely known about the fatalities. At a minimum, authors should provide some information about how the characteristics (age, gender, other variables known) of those who died by suicide in 2019 compared to those who attempted suicide that year, but were unsuccessful.

3. In the inclusion criteria section, authors state that all patients admitted to poison control centers with complete medical records were eligible for the study. Authors should provide information about poison control centers in Egypt so that readers understand who the population under study is. For example, are all cases of suspected poisoning referred to poison control centers (including deaths)?

4. In the Statistical Analysis section, authors should describe their approach for model building - how did they go from the univariate model to the multivariate model? Was overlap/collinearity between variables considered?

5. Univariate analysis section of results: Authors do not mention the finding for medium SES.

6. Multivariate analysis section of results: why do the authors combine categories for variables in the multivariate models? This needs to be justified and explained.

7. Authors spend several paragraphs in the Discussion about females attempting suicide, yet females are not significantly more likely than males to attempt suicide in the multivariable model.

Minor points:

1. Abstract: "The death rate due suicide" should be "The death rate due to suicide" and "which raise a serious concern" should be "which raises a serious concern". Also, to be consistent with later presented results in the manuscript, "CI95%" should be "95% CI"

2. Introduction: "among 15-29 year" should be "among people aged 15-29 years" and "2nd cause of death" should be "2nd leading cause of death"

3. Inclusion criteria: "For example, cut the wrists" is an incomplete sentence.

4. Data collection tools: "which was use in trend analysis" should be "which was used in trend analysis", "Confirmed by clinical evaluation..." is an incomplete sentence, and "age, gender, etc." should list all variables collected, not just saying "etc."

5. Statistical Analysis: "Qualitative data was" should be "Qualitative data were" and "Calculated by dividing..." is an incomplete sentence.

6. Figures 3 and 4: in the legends "slef-poisoning" should be "self-poisoning" and "self-posioning" should be "self-poisoning". Also, authors should clarify what is meant by total cases - is this attempted + successful?

7. Results, 1st paragraph: "attempted suicide by suicide" should be "attempted suicide by self-poisoning"

8. Table 1: the denominator written in the table for proportionate mortality rate does not seem correct, as this is the same as what is written for the self-poisoning suicide death rate.

9. Table 2 (and related text in Results): The percentages in the text of the paragraph at the bottom of page 7 (for aluminum phosphide and drugs used in attempted suicide) do not match what is presented in Table 2. Authors should also clarify in the text and the table that the percentages for the drugs are calculated out of the 467 individuals who used drugs.

10. bottom paragraph of page 7: "was pesticides (38.7%) then aluminum..." should be "was pesticides (38.7%), then aluminum..."

11. Univariate analysis results: the OR written for age<25 years does not match Table 3. Based on the numbers presented, it seems as though the number in the text is probably correct, while there is a typo in Table 3. Authors should carefully check the numbers and make the correction wherever it is needed. "female student" should just be "student" and the OR and CI for not working should be corrected (since the authors have re-written the OR and CI for student). "patients Those" should be "individuals" or "patients" and "confidence interval" should be "95% CI" to be consistent with other presented results in this section.

12. Discussion, 1st paragraph: "The rate of suicide attempts in the general..." should be "The rate of suicide attempts by self-poisoning in the general..." In this same paragraph, authors compare the result to a study of teens and make a point that the rate is much higher in 2019 than 2004. It is unclear why this comparison is relevant if the 2004 study is just about teens, while the current study is about all attempts.

13. Discussion, 2nd paragraph: "Suicides by self-poisoning reached 28.76" should be clarified. First, authors should state whether this is out of admissions or population (/1000 or /100000). Second, authors need to clarify if this is attempts, deaths, or attempts+deaths.

14. Last sentence on page 12 ("Examining several issues...") is an incomplete sentence.

15. Conclusion: authors do not mention financial reasons, which was more common than psychological reasons for the attempted suicide. Also, the summary sentence on the results from the logistic model needs to be rewritten, since the authors have combined variables together (students under the age of 25, for example), while the model looks at each variable while holding the other variables constant (so authors can say, "students, those under the age of 25 and those who live in a rural area..."

7. PLOS authors have the option to publish the peer review history of their article (what does this mean?). If published, this will include your full peer review and any attached files.

Reviewer #3: No

Reviewer #4: No

---

## [Author Response · Author response to Decision Letter 2]

18 Apr 2022

Response to reviewers’ comments

Thanks so much for your valuable comments that enriched the article. We responded to all comments and we hope you find it satisfactory and admired.

Reviewer #3: 

The authors have adequately addressed comments raised in a previous round of review and feel that this manuscript is now acceptable for publication except for multiple grammatical errors. I encourage authors to fix these before publication.

Suggestion for authors: can use online freely available software like Grammarly, etc.

We have sent the manuscript to a proofreader who had checked for any errors or mistake. However, we didn’t highlight the language changes as they were many, to avoid confusing the other responses to reviewer comments.

Reviewer #4: 

1. Abstract: authors present results from the univariate logistic regression models. However, the multivariate model is more appropriate for final results, since this accounts for other variables included in the model.

Corrected as per request. 

2. In the subjects and sampling section of the Methods, authors state that fatalities were excluded from the analyses of risk factors. Authors should provide additional justification for this decision. 

Totally agree with this point. Unfortunately, exclusion of fatalities from analysis was based on the reviewer requirement in the last round of revision. However, we had deleted the statement” fatalities were excluded from the analyses of risk factors”, and we included the fatalities in the analysis as revealed at Table (2). Besides, we elaborated in this part in the abstract, results, discussion, especially the point referring for significantly higher fatalities due to self-poisoning by Aluminum phosphide.

3. Some of the information collected on the attempts are likely known about the fatalities. At a minimum, authors should provide some information about how the characteristics (age, gender, other variables known) of those who died by suicide in 2019 compared to those who attempted suicide that year but were unsuccessful.

The authors, as per the request, conducted the test of significance to compare characteristics (age, gender, other variables known) of those who died by suicide in 2019 compared to those who attempted suicide that year. A table was added (Table 2). 

4. In the inclusion criteria section, authors state that all patients admitted to poison control centers with complete medical records were eligible for the study. Authors should provide information about poison control centers in Egypt so that readers understand who the population under study is. For example, are all cases of suspected poisoning referred to poison control centers (including deaths)?

Agree. This part was rephrased completely in the section of the methodology under the subheading Inclusion criteria. The route of admission to poison control centers and triaging the patients were mentioned in more comprehensive way. We hope it meets your expectations.

5. In the Statistical Analysis section, authors should describe their approach for model building - how did they go from the univariate model to the multivariate model? Was overlap/collinearity between variables considered?

This part was fixed and rephased with a reference. Based on Bursac et al., 2008, the best way to select the included varaibles is to subject all significant variables in the univariate logistic regression to multivariate logistic regression to detect the riskiest factors to attempted suicide by self-poisoning. 

6. Univariate analysis section of results: Authors do not mention the finding for medium SES.

Detailed results of univariate logistic regression, including the medium SES was reported in the results section as per request.

7. Multivariate analysis section of results: why do the authors combine categories for variables in the multivariate models? This needs to be justified and explained.

Based on that comment, and to allow simple and easy interpretation, we do uncombined all variables and re-performed logistic regression as revealed at Table (4) and the corresponding part in the results section.

8. Authors spend several paragraphs in the Discussion about females attempting suicide, yet females are not significantly more likely than males to attempt suicide in the multivariable model.

Right. The multivariate model didn’t outline being a female as one of the significant suicide attempt risk factors, as this model highlight the most significant predictor considering the other variables which showed the riskiest factors like SES, age, residency and occupation. However, we could not overlook the significance of gender seen in the univariate analysis and when comparing studied patients according to the successfulness and manner of exposure (attempted versus successful, and accidental versus suicide self-poisoning). 62% of the Self-poisoning attempters, and 79.6 were females (which constituted more than half of studied Patients. Moreover, the added paragraphs were response to the reviewer comments in the first and second rounds of revisions. 

the 

Minor points:

1. Abstract: "The death rate due suicide" should be "The death rate due to suicide" and "which raise a serious concern" should be "which raises a serious concern". Also, to be consistent with later presented results in the manuscript, "CI95%" should be "95% CI"

The Confidence interval 95% 

was fixed at the different sections of the text aligned with the results, as per request.

2. Introduction: "among 15-29 year" should be "among people aged 15-29 years" and "2nd cause of death" should be "2nd leading cause of death"

Fixed 

3. Inclusion criteria: "For example, cut the wrists" is an incomplete sentence.

Fixed

4. Data collection tools: "which was use in trend analysis" should be "which was used in trend analysis", "Confirmed by clinical evaluation..." is an incomplete sentence, and "age, gender, etc." should list all variables collected, not just saying "etc."

Fixed

5. Statistical Analysis: "Qualitative data was" should be "Qualitative data were" and "Calculated by dividing..." is an incomplete sentence.

Fixed

6. Figures 3 and 4: in the legends "slef-poisoning" should be "self-poisoning" and "self-poisoning" should be "self-poisoning". Also, authors should clarify what is meant by total cases - is this attempted + successful?

Fixed. The total number of suicides by self-poisoning reflects the sum of attempted and successful suicide by self-poisoning. This clarification was added to the mentioned figures’ ligands.

7. Results, 1st paragraph: "attempted suicide by suicide" should be "attempted suicide by self-poisoning"

Fixed

8. Table 1: the denominator written in the table for proportionate mortality rate does not seem correct, as this is the same as what is written for the self-poisoning suicide death rate.

Fixed, it was typing error.

9. Table 2 (and related text in Results): The percentages in the text of the paragraph at the bottom of page 7 (for aluminum phosphide and drugs used in attempted suicide) do not match what is presented in Table 2. Authors should also clarify in the text and the table that the percentages for the drugs are calculated out of the 467 individuals who used drugs.

Sorry for this mistake due to reanalysis of data based on reviewers’ comments, there was unintended mismatching between table and text which was figured out after submission. However, the numbers and percentages of Causative Agent used as mentioned at Table (2), is calculated out of 2724 as highlighted in the title of the table, and not out of 467. We used the word causative agent to include the different categories of used substance in terms of chemicals, drugs, pesticides,….etc. However, a brief hint was added to the text clarifying the overall number of cases included in the calculation.

10. bottom paragraph of page 7: "was pesticides (38.7%) then aluminum..." should be "was pesticides (38.7%), then aluminum..."

Fixed

11. Univariate analysis results: the OR written for age<25 years does not match Table 3. Based on the numbers presented, it seems as though the number in the text is probably correct, while there is a typo in Table 3. Authors should carefully check the numbers and make the correction wherever it is needed. "female student" should just be "student" and the OR and CI for not working should be corrected (since the authors have re-written the OR and CI for student). "patients Those" should be "individuals" or "patients" and "confidence interval" should be "95% CI" to be consistent with other presented results in this section.

The entire section was revised and corrected as per your suggestion.

12. Discussion, 1st paragraph: "The rate of suicide attempts in the general..." should be "The rate of suicide attempts by self-poisoning in the general..." In this same paragraph, authors compare the result to a study of teens and make a point that the rate is much higher in 2019 than 2004. It is unclear why this comparison is relevant if the 2004 study is just about teens, while the current study is about all attempts.

This paragraph was rephrased and fixed to alleviate the confusion. We fixed the mentioned words. Regrading specifying the teens, this was requested by the reviewer in the second round of revision, considering the significance of suicide by self-poisoning among the patients aged less than 25 in the current study. Moreover, we had added a reference supporting the findings of suicide by self-poisoning among teens. Then, in the next paragraph, we discussed the problem of suicides by self-poisoning among admissions without reflecting on specific age group. We hope that meets your expectations.

13. Discussion, 2nd paragraph: "Suicides by self-poisoning reached 28.76" should be clarified. First, authors should state whether this is out of admissions or population (/1000 or /100000). Second, authors need to clarify if this is attempts, deaths, or attempts+deaths. Fixed

14. Last sentence on page 12 ("Examining several issues...") is an incomplete sentence. Fixed

15. Conclusion: authors do not mention financial reasons, which was more common than psychological reasons for the attempted suicide. Also, the summary sentence on the results from the logistic model needs to be rewritten, since the authors have combined variables together (students under the age of 25, for example), while the model looks at each variable while holding the other variables constant (so authors can say, "students, those under the age of 25 and those who live in a rural area..."

Fixed

---

## [Decision Letter · Decision Letter 3]

9 May 2022

PONE-D-20-34054R3Trend and Epidemiology of Suicide attempts by Self-poisoning among EgyptiansPLOS ONE

Dear Dr. Kasemy,

Thank you for submitting your manuscript to PLOS ONE. After careful consideration, we feel that it has merit but does not fully meet PLOS ONE’s publication criteria as it currently stands. Therefore, we invite you to submit a revised version of the manuscript that addresses the points raised during the review process. Please submit your revised manuscript by June 23, 2022. If you will need more time than this to complete your revisions, please reply to this message or contact the journal office at plosone@plos.org. Please include the following items when submitting your revised manuscript:A rebuttal letter that responds to each point raised by the academic editor and reviewer(s). You should upload this letter as a separate file labeled 'Response to Reviewers'.A marked-up copy of your manuscript that highlights changes made to the original version. You should upload this as a separate file labeled 'Revised Manuscript with Track Changes'.An unmarked version of your revised paper without tracked changes. You should upload this as a separate file labeled 'Manuscript'.If applicable, we recommend that you deposit your laboratory protocols in protocols.io to enhance the reproducibility of your results. Protocols.io assigns your protocol its own identifier (DOI) so that it can be cited independently in the future. For instructions see: https://journals.plos.org/plosone/s/submission-guidelines#loc-laboratory-protocols. Additionally, PLOS ONE offers an option for publishing peer-reviewed Lab Protocol articles, which describe protocols hosted on protocols.io. Read more information on sharing protocols at https://plos.org/protocols?utm_medium=editorial-email&utm_source=authorletters&utm_campaign=protocols.

We look forward to receiving your revised manuscript.

Kind regards,

Saeed Ahmed, MD

Academic Editor

PLOS ONE

Journal Requirements:

**Comments to the Author**

1. If the authors have adequately addressed your comments raised in a previous round of review and you feel that this manuscript is now acceptable for publication, you may indicate that here to bypass the “Comments to the Author” section, enter your conflict of interest statement in the “Confidential to Editor” section, and submit your "Accept" recommendation.

Reviewer #3: All comments have been addressed

Reviewer #4: (No Response)

2. Is the manuscript technically sound, and do the data support the conclusions?

Reviewer #3: (No Response)

Reviewer #4: Yes

3. Has the statistical analysis been performed appropriately and rigorously? 

Reviewer #3: (No Response)

Reviewer #4: Yes

4. Have the authors made all data underlying the findings in their manuscript fully available?

Reviewer #3: Yes

Reviewer #4: Yes

5. Is the manuscript presented in an intelligible fashion and written in standard English?

Reviewer #3: Yes

Reviewer #4: Yes

6. Review Comments to the Author

Reviewer #3: (No Response)

Reviewer #4: The authors have addressed the majority of my earlier concerns. There just remain a few very minor points that should be corrected:

1. on page 8, 1st full paragraph beginning with "Out of the 2724 cases...": authors refer to female students less than 25 years old are at higher risk of death from suicide when referring to results in Table 2. The results would suggest that females have increased odds of death relative to males, students have higher odds of death than those working, and those <=25 have higher odds of death than those >25. So interpreting the results as the combination of those variables (female students less than 25) is not exactly consistent with the analysis. Authors should rephrase this to indicate that sex, occupation, and age are separate variables.

2. Also on page 8 in the Univariate analysis section, authors said in their response that they corrected the repeated OR and CI, but that is not reflected in the submitted clean or corrected version with changes highlighted. The OR for not working is written exactly the same as the OR for student, but Table 3 shows these are in fact different ORs and CIs. Authors should put the correct OR and CI for "not working" in the text.

3. In the 1st paragraph of the Discussion, authors added a percentage (37.4%) for the 1196 attempts made by individuals less than 25 - authors should recheck that percentage, since 1196/2523=47.4 and 1196/(2523+201)=43.9, so it is not clear where the 37.4% is coming from.

4. Table 1: authors should include a note about where the 1424 (corrected denominator for the proportionate mortality rate) comes from or what it refers to.

7. PLOS authors have the option to publish the peer review history of their article (what does this mean?). If published, this will include your full peer review and any attached files.

Reviewer #3: No

Reviewer #4: No

---

## [Author Response · Author response to Decision Letter 3]

10 May 2022

Response to reviewers’ comments

Reviewer #3 and Reviewer #4: 

Thanks so much for your support and great efforts to make our article presented in the best way.

Reviewer 4#: The authors have addressed the majority of my earlier concerns. There just remain a few very minor points that should be corrected:

1. on page 8, 1st full paragraph beginning with "Out of the 2724 cases...": authors refer to female students less than 25 years old are at higher risk of death from suicide when referring to results in Table 2. The results would suggest that females have increased odds of death relative to males, students have higher odds of death than those working, and those <=25 have higher odds of death than those >25. So interpreting the results as the combination of those variables (female students less than 25) is not exactly consistent with the analysis. Authors should rephrase this to indicate that sex, occupation, and age are separate variables.

Response: Thanks so much and we apologize for this misinterpretation. We corrected it as per your request on individual basis. Fortunately, this part was interpreted on individual basis on the discussion section as previously recommended.

2. Also on page 8 in the Univariate analysis section, authors said in their response that they corrected the repeated OR and CI, but that is not reflected in the submitted clean or corrected version with changes highlighted. The OR for not working is written exactly the same as the OR for student, but Table 3 shows these are in fact different ORs and CIs. Authors should put the correct OR and CI for "not working" in the text.

Response: Thanks so much and we apologize for this mistake (typo), and we corrected it. OR and CI were written separately for both students and not working 

3. In the 1st paragraph of the Discussion, authors added a percentage (37.4%) for the 1196 attempts made by individuals less than 25 - authors should recheck that percentage, since 1196/2523=47.4 and 1196/ (2523+201) =43.9, so it is not clear where the 37.4% is coming from.

Response: Thanks so much and we apologize for this typing error, and we corrected it from 37.4 to 47.4%

4. Table 1: authors should include a note about where the 1424 (corrected denominator for the proportionate mortality rate) comes from or what it refers to

Response: This part was fixed in the results section, and as a foot note under Table (1). 

The equation of proportionate mortality rate = number of deaths (suicide) due to self-poisoning divided by deaths from all causes. The total number of deaths during this year from all causes in the referred hospitals where the poising centers present equaled 1424.

Proportionate mortality rate: Deaths caused by a particular cause/Deaths from all causes

---

## [Decision Letter · Decision Letter 4]

3 Jun 2022

Trend and Epidemiology of Suicide attempts by Self-poisoning among Egyptians

PONE-D-20-34054R4

Dear Dr. Kasemy, 

We’re pleased to inform you that your manuscript has been judged scientifically suitable for publication and will be formally accepted for publication once it meets all outstanding technical requirements.

Kind regards,

Saeed Ahmed, M.D

Academic Editor

PLOS ONE

---

## [Editor Report · Acceptance letter]

7 Jun 2022

PONE-D-20-34054R4 

Trend and Epidemiology of Suicide attempts by Self-poisoning among Egyptians 

Dear Dr. Kasemy:

I'm pleased to inform you that your manuscript has been deemed suitable for publication in PLOS ONE. Congratulations! Your manuscript is now with our production department. 

Kind regards, 

on behalf of

Dr. Saeed Ahmed 

Academic Editor

PLOS ONE